# Design of Equity Crowdfunding in the Digital Age

Budi Agus Riswandi , Abdurrahman Alfaqiih * and Lucky Suryo Wicaksono

Faculty of Law, Universitas Islam Indonesia, Sleman 55584, Indonesia
* Correspondence: 094100401@uii.ac.id

**Abstract:** Equity crowdfunding is a form of alternative financing for MSMEs in Indonesia. However, the provision of equity crowdfunding still has various issues that boil down to the absence of guarantees of legal certainty for the parties. This, of course, can hinder the development of equity crowdfunding itself in the MSME financing scheme. For this reason, the review of this is carried out based on normative legal research, where it examines various applicable legal provisions in regulating equity crowdfunding. Studies are also based on statute, comparative and conceptual approaches. The result is that, first, the arrangement regarding equity crowdfunding has not provided guarantees of legal certainty for the parties; second, many countries develop equity crowdfunding regulatory frameworks that are oriented to guarantee legal certainty for the parties; and third, the design of equity crowdfunding arrangements that provide guarantees of legal certainty to the parties can be made in the form of co-regulation arrangements.

**Keywords:** equity crowdfunding; legal certainty; regulation

## 1. Background

The development of technology brings a breath of fresh air to the financial industry with the emergence of financial technology. Financial technology can be interpreted as the utilization of information technology developments to improve services in the financial industry. Another definition is the variety of business models and technological developments that have the potential to improve the financial services industry. One type of fintech that is quite the centre of attention these days is crowdfunding (International Organization of Securities Commissions 2017).

Crowdfunding is a relatively new phenomenon in the financial industry. Crowdfunding is touted as a revolutionary method of business funding because it can be an effective means to raise capital through the Internet media of a large group of people around the world to turn their ideas into a project or business (Dresner 2014, p. 3).

The rise of crowdfunding and its rapid expansion into the market were among the results of the global financial crisis. During economic crises such as the one that began in 2007–2008, it becomes increasingly difficult to find financial resources. Crowdfunding emerged as one solution, allowing businesses to overcome the obstacles of the global economic crisis (Müllerleile et al. 2014, p. 277). Crowdfunding creates new jobs and encourages economic recovery that breathes new businesses into starting businesses and at the same time motivates investors.

There are several types of crowdfunding in practice, namely donation-based crowdfunding, equity-based crowdfunding (Equity Crowdfunding), peer-to-peer lending, and reward-based crowdfunding. Some types of (Hossain and Oparaocha 2017, p. 4) crowdfunding have been commonly used in many countries, both in developed and developing countries. For example, in Indonesia, donation-based crowdfunding is used by kitabisa.com platforms; peer-to-peer lending has mushroomed in Indonesia and equity-based crowdfunding also exists, for example, in the *Santara* and *Crowdana* platforms.

The existence of equity crowdfunding (ECF) as a fintech can be run with the aim to increase financial inclusion for businesspeople. Data on the development of ECF in

developed countries such as South Korea—which began to be introduced in 2016—has managed to raise funds of USD 72 billion involving as many as 432 issuing companies.[1] In addition, a number of countries in Europe, particularly the UK, Germany, and France, have enacted regulations that do not restrict investors but rather protect and promote them, which has come to be called investor-protection-oriented policies. As a result, it is natural that these countries have a record high volume of investment from the public. However, there are also some other countries in Europe, such as Spain and Italy, that apply the opposite approach, namely the restrictive approach, wherein, through this approach, investors are over-protected, causing them to be reluctant to invest (Cicchiello et al. 2020).

The phenomenon of state policy responses to the rapid development of the crowd-funding industry indirectly, whether we like it or not, shows that the state has no other choice in responding to the crowdfunding industry but to respond to the challenges in the field of entrepreneurial finance proportionally and ensure protection for investors. This then does not rule out the possibility of giving rise to two forms of domestic regimes with the tightening and strengthening of regulations and financial regimes with an orientation to adjust to the needs of the crowdfunding market and local investors. Therefore, a country's government must be able to encourage the growth of crowdfunding by implementing integrity policies that cover the guarantee of internal market stability and facilitate the creation of competitive market conduciveness. Thus, it is hoped that the minimization of information asymmetry and increased transparency and investor protection can be created (Cicchiello and Leone 2020).

In Indonesia, ECF started to exist in early 2018; at that time, there was no regulation governing ECF in Indonesia. At that time, *Santara* had started an ECF business with ECF-based tokens or coins instead of stock or equity. At the end of 2018, regulations related to ECF in Indonesia by the Financial Services Authority were issued. OJK has issued regulations in the form of Financial Services Authority Regulation Number 37/POJK.04/2018 on *Urun Dana* Services Through Information-Technology-Based Stock Offerings (Equity Crowdfunding") ("POJK 37/2018"). In December 2020, OJK issued the latest regulation related to the Financial Services Authority Regulation No. 57/POJK.04/2020 on Securities Offerings through Information-Technology-Based *Urun Dana* Services ("POJK No. 57/2020").

Based on POJK No. 57/2020, in the *Urun Dana* service activities, there are three parties involved, namely the organizer, issuer, and financier. The organizer is an Indonesian legal entity that provides, manages, and operates the *Urun Dana* service. The issuer is an Indonesian legal entity in the form of a limited liability company that offers shares through the organizer. The financier is the party who purchases the issuer's shares through the organizer.[2]

Until now, there have been four *Urun Dana* service providers who have permission from OJK, namely *Santara, Bizshare, Crowddana,* and *Landx*. The *Santara* platform, managed by *PT Santara Daya Insiparatama*, is the first *Urun Dana* service platform and is the market leader in this industry. *Santara* is the first equity crowdfunding platform to receive permission from OJK based on Decree Number: KEP-59/D.04/2019.[3] Up until the fourth week of May 2020, *Santara* was recorded to have raised funds of more than IDR 45 billion, which has been distributed to 41 publishers. Up until the fourth week of May 2020, *Bizshare* is recorded to have raised funds of more than IDR 26 billion, which has been distributed to 26 publishers.[4]

In Indonesia, business actors are also dominated by micro, small, and medium enterprises (MSMEs), which also have enormous potential to be scaled up. Based on data on

---

1    https://republika.co.id/berita/q9bi9n370/dua-fintech-Urun-dana-ini-kerja-sama-dengan-ksei, accessed on 19 June 2020.
2    *Article 1 number (5) of POJK No. 57/POJK.04/2020 on Securities Offerings through Information-Technology-Based Fund Urun Service*, 2020.
3    https://duniafintech.com/santara/, accessed on 27 June 2020.
4    https://santara.co.id, accessed on 1 June 2020.

the number of MSMEs in 2020 in Indonesia, approximately 64 million, 41 million MSMEs are related to financing and banking institutions; beyond that, there are 23 million MSMEs who do not have access to funding.[5]

Obstacles for MSMEs to access capital from banking investment are the absence of collateral and the amount of interest on incriminating loans for MSMEs. This happens because the banking institution approach still uses a collateral-based approach instead of a project-based one. *Urun Dana*'s service still has the potential to be developed, so it can be one solution to overcome these obstacles.

With the platform, fund providers who have obtained permission from OJK will be able to spur on the development of *Urun Dana* services in Indonesia. The development of the ECF industry in Indonesia must be accompanied by protection for the parties in it, namely *Urun Dana* service providers and *Urun Dana* service users. What is meant as a user of the service *Urun Dana* is the issuer and financier. The issuer is an Indonesian business entity both in the form of legal entities and other business entities that issue securities through the *Urun Dana* service.[6] The financier is the party that purchases the issuing securities through the *Urun Dana* service.[7]

Protection of the parties in this fund service urgently needs to be realized considering that the securities crowdfunding industry or securities offering through the *Urun Dana* service is a relatively new industry and has many risks. This is in line with the provisions of Article 66 of POJK No. 57/2020, which requires the organizer and users to mitigate risks. What is meant by risk mitigation is the mitigation of all risks contained in the *Urun Dana* service, including business risk, investment loss risk, risk of a lack of liquidity, risk of the scarcity of dividend distribution, and risk of the dilution of share ownership.[8]

In addition to the risks mentioned above, there are still legal risks that are potentially faced by the parties. In terms of service providers, *Urun Dana* has a very dominant role in the business process of *Urun Dana* services. The organizer is a provider of *Urun Dana* service platform funds that are used as a meeting place for publishers and financiers of electronic funds. In addition, the organizer is also a party who has the authority to conduct an assessment and review of the eligibility of prospective issuers. Considering this, service organizers should have the same guidelines for assessing and reviewing prospective publishers.

Guidelines on conducting assessments and the assessment of prospective publishers conducted by the organizer should also apply the precautionary principle. Financiers in the service of *Urun Dana* only contribute to the information provided by the publisher and published by the organizer through its platform. So, it can be said that the organizer is the party that provides all the information that is the main consideration for the financier to determine his investment decision. As such, the operator has a responsibility to provide and/or provide up-to-date information about *Urun Dana* services that is accurate, honest, clear, and not misleading.[9]

From the financier's side, more risks are faced. Starting from the risk of investment losses, there is the risk of a lack of liquidity, the risk of the scarcity of dividend distribution, and the risk of the dilution of share ownership. These risks can be minimized if the organizer can apply the precautionary principle in conducting the assessment and review of prospective issuers to ensure that securities offered by the publisher through

---

[5]    https://bisnis.tempo.co/read/1336881/jokowi-minta-23-juta-umkm-diberi-bantuan-pembiayaan-modal-kerja/full&view=ok, accessed on 18 June 2020.

[6]    *Article 1 number 6 of Financial Services Authority Regulation No. 57/POJK.04/2020 on Securities Offerings Through Information-Technology-Based Fund Urun Services*, 2020; *Article 1 number 7 of Financial Services Authority Regulation No. 57/POJK.04/2020 on Securities Offerings Through Information-Technology-Based Fund Urun Services*, 2020.

[7]    *Article 1 number 8 of Financial Services Authority Regulation No. 57/POJK.04/2020 on Securities Offerings Through Information-Technology-Based Fund Urun Services*, 2020.

[8]    *Explanation of Article 66 of Financial Services Authority Regulation No. 57/POJK.04/2020 on Securities Offerings Through Information-Technology-Based Fund Urun Services*, 2020.

[9]    *Article 73 POJK No. 57/POJK.04/2020 on Securities Offerings through Information-Technology-Based Fund Urun Service*, 2020.

the subscription platform are of good quality and carry minimal risk when purchased by financiers.

In terms of the publisher, the most dominant risk is business risk. This happens considering that the issuer who will provide the securities offering through this *Urun Dana* service is a micro, small, and medium enterprise (MSME). MSMEs are still vulnerable to mitigating business risks, one of which maintains the company's ongoing concerns. Limitations in terms of capital and management represent one factor, and innovation is usually a contributing factor.

It is important that security and protection guarantees to the parties in this service are strengthened. This is an effort to keep the fintech service industry sustainable and not detrimental to the parties. The regulator whose nature is OJK should take a role in creating guarantees, and protections for the parties can be accommodated by providing sufficient regulation to provide legal guarantees and protections for the parties as well as the enforcement of regulations by the Financial Services Authority.

Referring to the current regulations, namely POJK 57/2020, there are still some loopholes that should be perfected to create security guarantees and protection of the parties in the offering of securities through the *Urun Dana* service. Therefore, this paper wants to research "The Design of Equity Crowdfunding Arrangements That Guarantee Legal Certainty in The Digital age".

## 2. Problem Formulation

Based on the background of the problems described above, it can be formulated as follows:

1. Has Indonesia's equity crowdfunding arrangement provided a guarantee of legal certainty in the digital age?
2. How is the development of equity crowdfunding arrangements in some countries in the digital era?
3. How is the design of Indonesia's equity crowdfunding arrangement that can provide guaranteed legal certainty in the digital era?

## 3. Library Review

Research on the topic of equity crowdfunding in Indonesia has been widely carried out. The following are some studies related to equity crowdfunding:

The first study, entitled "Urgency of Implementing Regulatory Sandbox by Financial Services Authority as An Effort to Protect Law for Equity Crowdfunding Financiers", written by Na'im Fajarul Husna, *Universitas Sebelas Maret*, was published in *Journal de Jure Volume 12 Number 1* in April 2020. This study discusses equity crowdfunding practices in Indonesia and compares regulatory sandboxing in other countries. The results of this study concluded that existing regulations have not accommodated legal protections for financiers and that potential losses can be incurred by organizers and issuers. Therefore, to prevent losses from being incurred, there must be a trial to ensure the readiness of the organizer. A regulatory sandbox can be a means to trial regulations for prospective equity crowdfunding companies. Later, it is expected that the trial period in the regulatory sandbox can be a benchmark of whether the company is viable and allowed to operate in equity crowdfunding. This needs to be carried out to provide legal protection for equity crowdfunding financiers.

The second study, entitled "problematical Legal Protection of The Parties in The Transaction of *Urunan Dana* Services Approved the Information-Technology-Based Stock Offering (Equity Based Crowdfunding)", was written by Viodi Childnadi Widodo and Dona Budi Kharisma, *Sebelas Maret University*, published in the *Journal of Private Law Volume VIII Number* 2 in December 2020. This research aimed to find out the form of legal protection for the parties and the problems faced by the Financial Services Authority in providing legal protection efforts to the parties in equity crowdfunding services. The results of this study concluded that legal protection in equity crowdfunding activities has not been optimal,

which is due to the problems faced by OJK in providing legal protection efforts to the parties in the form of (a) legal substance in the form of lack of detail based on the Capital Market Law, as further affirmed in the Financial Services Authority Act, which should be the reference of POJK Number 37/POJK.04/2018, which lacks detail and only includes sections on public offering and securities trading activities as stipulated in the Capital Markets Act; (b) the legal structure in the form of the organizer's platform, which provides less clarity on risk mitigation if there is a risk that it harms the financier or issuer; and (c) a legal culture in the form of a society that is less concerned about the importance of information about the form of legal protection to users.

The third study, entitled "Legal Protection of Users of *Urun* Dana Services Through Information-Technology-Based Stock Offering", written by Suriyadi, Alauddin State Islamic University, was published in *The Journal of Panorama Law Volume 5 Number 2* in December 2020. This research discusses the extent to which POJK regulation No. 1/POJK.07/2013 on consumer protection of the financial services sector and POJK Number 37 of 2018 provides protection for parties in information-technology-based fund-based services. The results of this study concluded that, based on both OJK regulations mentioned, the responsibility for user losses (financiers) due to the error of directors, employees, and/or third parties working for the organizer is the responsibility of the organizer, even though the provisions of Article 1365 of the Civil Code regulate that the party who made the mistake must be responsible for the loss; as such, the OJK regulation should regulate more details regarding responsibility based on errors and whose actions result in losses, so ensure that it is not due to the actions of third parties who cooperate with the organizers and make mistakes and acts of violation that result in the loss of the financier of full responsibility to the organizer.

The research conducted by this author is different from research that has been carried out before. The difference lies in the previous studies that have been outlined above, still limited to looking at problems or legal problems related to the implementation of information-technology-based stock offerings conducted by the parties, who in this case are organizers, issuers, and financiers. The emerging regulations are associated with the extent to which applicable laws and regulations provide legal protection for the parties. In addition, the regulations used in previous legal research still use the basis of POJK No. 37 of 2018, which has now been revoked and declared invalid since the enactment of POJK No. 57 of 2020 junto POJK No. 16 of 2021. This research not only looks at problems in the equity crowdfunding industry but more deeply tries to provide a better design arrangement to provide certainty for the parties in information-technology-based fund-based services in Indonesia.

## 4. Theory of Legal Certainty

In a society, there is always a legal aspect (*ubisocietas*, *ibi ius*). The law is an inseparable aspect of society. Laws exist to ensure the order that the forces that control society want to implement. The object is uniformity of action so that one member of the public can know how, under certain circumstances, others tend to behave; this is the essence of security (Wade 1941).

Law in this context serves to create public order. The order of society that the law wants to create should aim at justice in the enactment of the law. Justice of the law should be able to contain two things, namely (1) that there is legal stability and (2) that this legal stability can provide a measure of legal protection for all its members. Justice in law is a quality that cannot be explained except by reference to a further value, and that value is determined by whatever general opinion occurs in society at the time. From this, it can be understood that legal justice is realized in the form of values that are clearly and logically formulated; the value formulated must still refer to the values that became the general opinion of the people at that time (Wade 1941).

From this understanding, legal justice is something based on dynamic values. This condition certainly means that legal justice is often considered contrary to legal certainty as another legal purpose. Legal justice and legal certainty are two things that cannot be

separated. The current concept of legal certainty does not stop only at the availability of a stable value, but that value, when it becomes a decision, must be acceptable to society at that time.

This is in line with that stated by Aulis Aarnio and Alexander Peczenik, who distinguish between formal and substantive legal certainties. Formal legal certainty implies that the law, including court rulings, must be predictable. The law must meet the requirements of clarity and stability so that those concerned can accurately calculate the legal consequences of their actions as well as the outcome of the legal process. Furthermore, substantive legal certainty relates to the rational acceptability of legal decision-making. In this sense, it is not enough that laws and rulings are predictable: they must also be accepted by the legal community concerned (Paunio 2009).

With the existence of two legal certainty requirements based on the clarity of requirements and acceptance from the legal community concerned, this has demanded legal certainty viewed from two sides. The two sides are, on the one hand, that legal certainty requires decisions that are consistent with the framework of the existing legal system; on the other hand, the demand for legitimacy demands decisions that are not only consistent in relation to the surrounding legal system but must also be rationally justified for all participants to accept them as rational decisions (Paunio 2009, p. 2).

In line with the above, R. Lanneau argues that legal certainty also depends on the interrelationship between facts and the law. There are three dimensions associated with legal certainty. First, there is certainty about the content of the law itself, namely the certainty of legal material. Second, there is the certainty of transition from fact to law, or in other words, the legal certainty of the event. Third, there is the existence of certainty in forming a circle, associating the law with facts and forms of the efficacy of legal certainty (Al-Fatih and Aditya 2019).

In its implementation, legal certainty in the form of laws and regulations is not seen by everyone with sufficient knowledge and expertise, thus having implications for great uncertainty. On this basis, it becomes an obligation of the legislative body to consider the special characteristics of the laws and regulations that will be established. In addition, the legislative body must also be able to create a "safety net" for each group that will be covered through the laws and regulations it establishes (ter Borg and Stoter 2010).

The lack of standards in the law will be compensated by the regulator's interpretation of how targets should be met, creating their own standards. One possible consequence is that individuals will be confronted ex post facto with the standards of the supervisory authority, which may reflect internal guidelines on how their officials should perform their supervisory functions. Further consequences of this kind of practice will cause two serious impacts, namely: (1) it will cause uncertainty for groups covered by the legislation and (2) undermine the desired purpose of the legislation itself. Thus, in order for goal-based regulations to be effective, groups covered by the legislation must enjoy freedom and responsibility, especially in terms of the choice of ways to achieve regulatory compliance (ter Borg and Stoter 2010, p. 3).

In the context of oversight, the legislature can maintain legal certainty by imposing an obligation to supervisory authorities to draw up policy rules regarding the details of surveillance. Everyone should be able to be involved when these policy rules are made, so that, in this way, they can have their own input on their's influence and responsibilities concerning the details of the standards. Through this, one can exchange knowledge and expertise with others covered by standards and supervisory authorities. With this pattern, one will tend to support the results of formulating standard details. Furthermore, the laws and details of the standard can be clearly informed to everyone to advance legal certainty (ter Borg and Stoter 2010, p. 6).

## 5. Legal Theory as a Tool of Social Engineering

The thought of law as social engineering emerged from Roscou Pound and was later accepted by some legal experts including in Indonesia. The term 'social engineering'

intentionally places more importance on the state than the interests of the individual. In Indonesia, this legal thought was first adopted by Mochtar Kusumaatmadja and referred to as the theory of development law. Furthermore, this thought was also embraced by Satjipto Rahardjo. Roscou Pound argues that laws should be developed based on their philosophy of usefulness. He argued that the law should be able to meet the needs of society in the modern century (Martin 1965). So, he posited: "law in action, not law in the books", in view that the law be used as an instrument of social engineering (social engineering) to secure social change. The construction of law as social engineering is a form of understanding that law is part of social change. Social change is often referred to as the dynamics of society or social transformation as a definite thing (Gray 1999, p. 672). Thus, it can be understood that the law has a strong relationship with social reality and that there must by a symbiotic relationship between the two. Lawrence Stone's book *Road to Divorce* is described by Ron Shaham as claiming that the legal relationship with the public or society is simple. According to him, the law can be influenced by society, or on the other side of the law that affects society. First, the law is influenced by society, one of which is indicated by the existence of legislators who make it so that their own laws are influenced by the values of the time in which the law is made. Second, on the other hand, it is very different from what it should be; the law itself is a form of public opinion and public behaviour (Ishak 2013). The existence of the law occurs in two ways, namely departing from the community and then being interpreted by lawyers and judges or being interpreted by judges and lawyers and applied in the community (Shaham 1995). Thus, it is very clear that there is a reciprocal relationship between established laws, theories of justice, and the background of social, economic, and cultural conditions. Based on the above explanation, it is known that the law can be interpreted simply, namely bottom-up (from the social reality of society to the law of the state) or top-down (from law to the reality of society). So, it can be understood that the position of law as social engineering lies in the second type, namely the law that governs society. If the law originates from the institution of state law, then the interests of the state are what govern what happens in society. Law as social engineering is a phenomenon that emerged in the 20th century. The emphasis of the meaning of social engineering, in this case, is that the law is a body of social rules that are already embedded in society and have become a condition of political decisions. The law, then, becomes a means of political implementation that has lost its roots in traditional life; the law does not see the past but looks forward by looking at the future that is aspired to. Thus, the law no longer maintains the status quo but instead makes social changes (Rosana 2013).

Law as social engineering (social engineering) is an effort to deal with legal problems that occur when certain laws formed and applied cannot function properly. Some obstacles usually exist. The symptoms of the obstacles in the field of law can arise from law enforcement, justice seekers, and other groups in the community. Thus, the law is a means of social engineering aimed at changing the activities of citizens, in accordance with the legal objectives that have been set (Soekanto 2013, p. 135).

## 6. Regulation and the Internet

Economic globalization has encouraged the globalization of law. This phenomenon is accelerated when economic and legal globalization is facilitated using information technology, especially the Internet. The use of Internet technology is associated with patterns in the interaction between people, both individually, organizationally, and in government, which have given birth to new phenomena and characteristics in the field of law.[10]

---

[10] Jones (2000): The Internet can be defined as a network of interconnecting networks—local, regional and global connected by the user via telephone or satellite connection. Its development was started in the early 1960s by a young group working at the Advance Research Project Agency (ARPA). The development was originally carried out as an experiment of the United States military's defense computer network known as ARPAnet. In the 1980s it was developed by a U.S. National Science Foundation for local and regional defense purposes. In the 1990s, it began to expand widely into the human life sector. See also Segura-Serrano (2006). In this context, there are two opposing groups on the importance of the law regulating problems on the Internet.

New phenomena and characteristics in the field of law not only concern the legal relationships carried out between humans no longer being face to face but also the space and time no longer existing between territorial boundaries, where the legal relationship can be performed by anyone and at any time by crossing the borders of the country (extraterritorial). The presence of phenomena and legal characteristics as above has given birth to a new challenge. Nicola Lucchi stated that the information technology revolution and digitalization of contacts have resulted in many new possibilities and challenges (Lucchi n.d.).

The Internet is a global multi-communication space wherein information and communication technologies converge and regulations are multi-layered. The basic unit of Internet regulation is the code, programming software, or logic that makes the Internet function. In his highly acclaimed book *Code and Other Laws of Cyberspace,* Lawrence Lessig (1999) conceptualized Internet regulation as a space dominated by corporate commercial technology and the underlying logic of computer code, operating within the framework of the rule of law. Indeed, by creating protocols such as those designed to protect children from harmful content on the Internet and by setting standards for anything from HTML to peer-to-peer music exchanges, the Internet community clearly regulates the Internet (Eko 2013).

In addition, through code, information and communication technology is used to regulate the behaviour of Internet users on a global scale, in accordance with the values, ideals, and ethics of hardware, controllers, and designers of hardware and software. In addition, regulators around the world use architectural, software, and hardware solutions to address cybercrime, network security, reliability, privacy, and intellectual property issues.

In line with the use of code to regulate the behaviour of Internet users, the ethical aspect of netiquette becomes another means that can control the utilization of the Internet that is not in harmony with the ethics of Internet users themselves.

Pręgowski stated that netiquette is an obligation that is present by users acting when connected to the Internet. Just like the rules of ethics in the real world, netiquette also encourages users to adhere to ethical and moral rules—often unwritten—to create a comfortable, peaceful, and peaceful common space (Fahrimal 2018).

The market aspect is the next instrument in controlling the utilization of the Internet from all forms of abuse that can harm its users. The role of the market in controlling the utilization of the Internet is realized in the form of a market review of a digital service. With this digital service review, it has had an impact on the behaviour of service providers to always provide optimal services to Internet users.

The use of the Internet that can be controlled strongly; in fact, it is still very dependent on regulations made and strengthened by the government. Since the Internet became a global multi-communication medium in the 1990s, countries around the world have enacted laws aimed at regulating segments of the Internet within their jurisdictions or at least brought parts of that network into the scope of their systems and legal jurisdictions. The result is a set of Internet regulatory models at the international and national level that govern everything from Internet infrastructure to social networking sites such as Facebook, Myspace, and Twitter network organizations through communication technology; P2P network (Eko 2013).

In its development, this regulation was not only born by the government but also developed by Internet users through self-regulation. Self-regulation is a set of provisions made by the parties themselves to regulate the utilization of the Internet. The real form of self-regulation can be a term and condition compiled by one party then agreed upon by the other party.

---

The first group is supported by libertarians, where they assume that the Internet is lawless and should not be governed by law; the second group is supported by traditionalists who consider that the state is the most viable political and legal institution to regulate the Internet. In addition to these two groups, there is a third group that combines the two groups. This group is called mixed regulation and governance and wants efforts to combine national laws and regulations that apply in the community.

## 7. Method

Research is normative legal research that examines the law related to equity crowdfunding as a rule (Abdurrahman 2009).[11] In addition, this type of legal research is carried out by researching secondary data using deductive thinking methods. The approach used in this research uses a statutory approach, conceptual approach, and comparative approach. Analysis of data in this study by means of qualitative descriptive analysis is carried out by collecting legal material related to the problem and then ensuring sentences are clear, orderly, logical, and effective so that a clear picture is obtained precisely and conclusions can be drawn (Asikin 2006). This study includes data classification activities, editing, the presentation of analytical results in the form of narratives, and decision-making.

## 8. Indonesia's Equity Crowdfunding Arrangement Has Provided Guaranteed Legal Certainty in the Digital Age

The term 'crowdfunding' is a derivative of the more popular term 'crowdsourcing', which describes the process of outsourcing a job to a group of people (the Internet community) and relying on their assets, resources, knowledge, or expertise. If compared with the concept of Hemer (2011, p. 8), crowdfunding is more intended to obtain funds. Funds raised for crowdfunding can be used for various purposes such as the completion of certain projects, donations of humanitarian activities, and others (Josua et al. 2019).

The concept of crowdfunding is an alternative form of funding to traditional loans/financing. Crowdfunding is open to all private people as well as economic actors or groups of people who provide small or large grants. Basically, crowdfunding still refers to funding one project/business but by involving the process of new media, namely the Internet itself.

Crowdfunding activities have various types. Until now, at least the crowdfunding model could be divided into four types, including donation-based crowdfunding, reward-based crowdfunding, lending-based crowding, and equity-based crowdfunding (Gabison 2015, p. 3). Equity-based crowdfunding is the type of crowdfunding that most attracted the attention of regulators in various countries; this is because equity-based crowdfunding involves selling shares of a company to the public, where usually similar activities have their own regulations such as regulations related to capital markets. As a result, many regulators in various countries have begun to review and design regulations or have even issued special regulations related to equity-based crowdfunding (Ahlers et al. 2013, p. 8). In the United Kingdom, regulations regarding equity crowdfunding have been enacted since March 2014, while in the United States (Financial Conduct Authority Policy Statement n.d., p. 5), crowdfunding arrangements have also been regulated since 2012 with the enactment of the JOBS Act's Regulation.[12]

Regulation of the Financial Services Authority No. 37 of 2018 on *Urun Dana* Services Through Information-Technology-Based Stock Offerings ("POJK No. 37/2018"). POJK No. 37 of 2018 has been revoked by the Law of the Financial Services Authority Regulation No. 57 of 2020 on Securities Offerings through Information-Technology-Based *Urun Dana* Services ("POJK No. 57/2020"). Recently, POJK No. 57/2020 was amended again with the promulging of Financial Services Authority Regulation No. 16/POJK.04/2021 on Changes to Financial Services Authority Regulation No. 57/POJK.04/2020 on Securities Offerings Through Information-Technology-Based *Urun Dana* Services ("POJK No. 16/2021"). POJK No.16/2021 only changes some articles in POJK No. 57/2020. Regulation in POJK No. 57/2020 remains valid except for some articles that are amended with POJK No.16/2021.

For the Indonesian context, the service industry has begun to develop in Indonesia since the enactment of the Financial Services Authority Regulation No. 37 of 2018

---

11　According to Abdurrahman (2009), normative legal research is a research that conceptualizes the law as what is written in the law in books or law as a rule or norm as a benchmark of human behaviour that is considered appropriate.

12　https://www.sec.gov/divisions/marketreg/tmjobsact-crowdfundingintermediariesfaq.html, accessed on 15 November 2021.

on *Urun Dana* Services Through Information-Technology-Based Stock Offerings ("POJK No. 37/2018"). POJK No. 37/2018 is the only legal umbrella governing *Urun Dana* services. Based on POJK, there are three parties involved in the *Urun Dana* service, namely: the arrangement of funds in POJK is certainly far from perfect because this POJK still needs to be "tested" for whether it can be an adequate legal umbrella for the service industry, which is still very new.[13]

POJK 37/2018 was then changed in 2020 with the enactment of changes to POJK No. 37/2018. POJK No. 37 of 2018 was revised on 10 December 2020, with the Financial Services Authority Regulation No. 57 of 2020 on Securities Offerings through Information-Technology-Based *Urun Dana* Services ("POJK No. 57/2020"). The main reason for the change to POJK No. 37/2018 was because, in its development, ECF is considered unable to meet the financing needs of small and medium enterprises and start-up companies (start-up companies) considering that the form of legal entities that underlie the establishment of SMEs is not entirely in the form of limited liability companies so POJK 37/2018 cannot be utilized optimally by SMEs as one of the sources of funding for SMEs.[14]

POJK 57/2020 is expected to expand the scope of the issuer in *Urun Dana* services and expand the scope of the issuer in *Urun Dana* service so that it can include offering securities other than equity securities in the form of shares; as such, it is necessary to replace POJK 37/2018. Based on POJK No. 57/2020, the securities offered through this *Urun Dana* service are no longer limited to shares but have been expanded with securities. The term 'securities' refers to debt recognition letters, commercial securities, stocks, bonds, debt proof marks, units of participation of collective investment contracts, futures contracts on securities, and any derivatives of securities.[15]

In 2021, OJK issued Financial Services Authority Regulation No. 16/POJK.04/2021 on Changes to Financial Services Authority Regulation No. 57/POJK.04.2021 on Securities Offerings Through Information-Technology-Based *Urun Dana* Services ("POJK No. 16/2021"). The cause of the promulgation of POJK No. 16/2021 was that there are problems with fulfilling licensing requirements as a *Urun Dana* service operator due to differences in applicable arrangements, especially related to the obligation to implement a private scope electronic system to register with the ministry that subscribes to government affairs in the field of communication and informatics. In connection with this, to harmonize the differences in the policy of the regulation, it is necessary to adjust by perfecting the material content of the arrangement in POJK regulation No.57/2020.[16]

Substantively, POJK, which regulates *Urun Dana* services, regulates the criteria of parties related to the service. The parties are (Hartanto 2020):

1. Publisher

The issuer is an Indonesian legal entity in the form of a limited liability company that offers shares through the organizer. The issuer must be a limited liability company considering that the legal entity in Indonesia that can issue shares is a limited liability company.[17] The issuer is not a public company as referred to in Law No. 8 of 1995 on

---

13   *BOY No. 37/2018 dan BOY No. 57/2020*, n.d., n. Urun Dana service is a term used in Indonesia to refer to Crowdfunding. The term 'Urun Dana service' is used in POJK No. 37/2018 and POJK No. 57/2020. In POJK No. 37/2018 known as stock offering through Urun Dana service while in POJK No. 57/2020 used the term securities offering through the Urun Dana service.

14   *General Explanation of Financial Services Authority Regulation No. 57 of 2020 on Securities Offerings through Information-Technology-Based Fund Urun Services*, 2020.

15   *Article 1 paragraph 2 POJK No. 57/2020*, n.a.

16   *General Explanation of Financial Services Authority Regulation No. 16/POJK.04/2021 on Changes to Financial Services Authority Regulation No. 57/POJK.04.2021 on Securities Offerings Through Information-Technology-Based Fund Urun Services*, 2021.

17   *Article 1 number 6 POJK No. 37 /POJK.04/2018 on Urun Dana Services Through Information Technology-Based Stock Offering (Equity Crowdfunding)*, n.a.

Capital Markets if the number of shareholders of the issuer is not more than 300 parties and the amount of the issuer's paid-up capital is not more than IDR 30,000,000,000.00.[18]

Based on data from December 2020, the number of UMK issuers that have registered in four new subscriptions amounted to 176 publishers, with the amount of fundraising being worth IDR 362 billion.[19] When compared to the number of MSMEs from government data that reached 64 million business actors, the contribution is still very small.[20]

2.  Organizers

Equity crowdfunding organizers are Indonesian legal entities in the form of limited liability companies or cooperatives. The limited liability company can be a securities company that has obtained approval from[21] the Financial Services Authority to carry out other activities as an organizer. The organizers who are incorporated cooperatives are limited to the type of service cooperative.[22]

Organizers in the form of limited liability companies and cooperatives must have paid-up capital of at least IDR 2,500,000,000 at the time of applying for a permit.[23] Organizers who will conduct the *Urun Dana* service (equity crowdfunding) must have a business license from the Financial Services Authority.[24]

Until now, there have been seven fund service providers who have obtained permission from OJK, namely PT Santara Daya Inspiratama (Santara), PT Investasi Digital Nusantara (Bizhare), PT Crowddana Teknologi Indonusa (CrowdDana), PT Numex Teknologi Indonesia (LandX), PT Dana Saham Bersama (Dana Saham), PT Dana Investasi Bersama (FundEx), and PT Syafiq Digital Indonesia (Shafiq).[25] All the organizers have also joined the Urun Dana Indonesia Service Association (ALUDI), which was endorsed by the Ministry of Law and Human Rights on 2 November 2020.[26]

3.  Financiers

The financier is the party who purchases the issuer's shares through the organizer. Financiers can be individuals or legal entities. Especially for individual financiers who do not have experience investing in the capital market, as evidenced by the ownership of securities accounts at least 2 years before the stock offering, it must meet the qualifications as stipulated in Article 42 of POJK No. 37/POJK.04/2018 concerning *Urun Dana* Services Through Information-Technology-Based Stock Offerings (Equity Crowdfunding), namely:[27]

a.  Each financier with an income of up to IDR 500,000,000.00 (five hundred million rupiah) per year can buy shares through the *Urun Dana* service amounting to at most 5% (five percent) of income per year; and

18  *Article 6 POJK No. 37 /POJK.04/2018 on Urun Dana Services Through Information Technology-Based Stock Offering (Equity Crowdfunding)*, n.a.

19  https://investor.id/market-and-corporate/267572/equity-crowdfunding-himpun-dana-rp-365-miliar, accessed on 6 October 2021.

20  https://finansial.bisnis.com/read/20211018/563/1455624/potensi-belum-tergarap-optimal-pemain-fintech-Urun-dana-bakal-makin-ramai, accessed on 26 October 2021.

21  *Article 10 POJK No. 37 /POJK.04/2018 on Urun Dana Services Through Information-Technology-Based Stock Offering (Equity Crowdfunding)*, 2018.

22  *Article 11 POJK No. 37 /POJK.04/2018 on Urun Dana Services Through Information-Technology-Based Stock Offering (Equity Crowdfunding)*, 2018.

23  *Article 12 POJK No. 37 /POJK.04/2018 on Urun Dana Services Through Information-Technology-Based Stock Offering (Equity Crowdfunding)*, 2018.

24  *Article 7 POJK No. 37 /POJK.04/2018 on Urun Dana Services Through Information-Technology-Based Stock Offering (Equity Crowdfunding)*, 2018.

25  https://finansial.bisnis.com/read/20211018/563/1455624/potensi-belum-tergarap-optimal-pemain-fintech-Urun-dana-bakal-makin-ramai, accessed on 23 December 2020.

26  https://finansial.bisnis.com/read/20201215/563/1331110/resmi-fintech-Urun-dana-atau-equity-crowdfunding-kini-punya-asosiasi, accessed on 25 December 2020.

27  *Article 1 number 7 POJK No. 37 /POJK.04/2018 on Urun Dana Services Through Information-Technology-Based Stock Offering (Equity Crowdfunding)*, 2018.

b.  Any financier with an income of more than IDR 500,000,000.00 (five hundred million rupiah) per year can buy shares through the *Urun Dana* service amounting to at most 10% (ten percent) of income per year.

Based on data as of October 2021, there was an increase in the number of financiers from December 2020 data; the number of financiers increased from 22,341 financiers to as many as 34,674 financiers.[28]

POJK No. 57/2020 *Junto* POJK No. 16/2021 as the only legal payment of *Urun Dana* service arrangements in Indonesia regulates the principals that include general provisions, *Urun Dana* service providers, *Urun Dana* services, users of *Urun Dana* services, *Urun Dana* service agreements, risk mitigation, education, and protection of users of *Urun Dana* services, principles of knowing customers, other provisions, administrative sanctions, transitional provisions, and the closing provisions. Whether the existing rules that have guaranteed the legal certainty of the parties is unclear.

Looking at the changes in the regulation of POJK regulations on SCF, it shows that the SCF industry is still looking for an ideal arrangement that can support the development of this industry both in terms of increasing the number of issuers or financiers who invest in SCF. However, what needs to be considered is that the ideal regulation should be able to provide legal certainty for organizers, publishers, and financiers in the SCF industry.

The SCF industry has also experienced many challenges not only from the regulatory aspect of wiring but also from various other aspects including the challenge from the publisher side to introduce a new type of investment for the wider community, the challenge for the organizer to provide a quality publisher for financiers, the challenge for the publisher to conduct a good and responsible review, and the challenge for the publisher to maintain the trust of the financier.

The challenge for the organizer to provide a quality publisher certainly requires human resources owned by the organizer. Considering that POJK 57/2020 has not regulated the obligation to carry out research related to the financial and legal side of the issuer by an independent profession that already has the required compensation, in this case, the publisher must ensure that the internal human resources possessed have enough competence to conduct a review of prospective publishers so that the data submitted in the publisher's prospectus are valid. The correctness of the information contained in the prospectus will certainly be the main factor in the investment decision of financiers.

Other challenges are also faced by publishers to maintain the going concern of companies that are funded by financiers. For MSME companies, maintaining the company's ongoing concerns is a tough challenge. Reflecting on the slowing economic situation due to the pandemic and the internal management of MSME who sometimes do not have the ability to develop/scale-up companies funded by financiers is necessary.

Maintaining the trust of financiers, especially in implementing the principle of openness in the management of companies funded by financiers, is another challenge for publishers. However, when compared to the obligation to implement information disclosure by open companies in the capital market, the obligation of information firmness for issuers should not be too difficult to fulfill. It turns out that there are still many publishers who do not carry out the disclosure of information to financiers, and no sanctions are imposed for publishers who violate the terms of disclosure of such information.

In terms of financiers, there are also challenges with the expanded scope of securities traded through SCF also needing to be anticipated by financiers by more carefully having securities instruments that will be invested in. Each type of investment instrument has different investment risks. Financiers who invest in SCF have high risks, including[29]

a.  Effort;
b.  Investment;

---

28  https://investor.id/market-and-corporate/267572/equity-crowdfunding-himpun-dana-rp-365-miliar, accessed on 26 October 2021.

29  *Article 16 paragraph (1) letter i POJK No. 57/2020*, n.a.

c. Liquidity;

d. electronic system failure;

e. scarcity of dividend distribution, and/or dilution of share ownership, if the issued securities are shares and default on debt securities;

f. Sukuk, if the issued effect is a debt or the Sukuk Effect.

When viewed from the description of the type of risk in the SCF above, the financier is faced with many risks. To commit these risks, financiers must be more careful in determining the effects and adjusting to their respective risk profiles.

In terms of financiers, investment through SCF is a type of investment that belongs to the head of investment that is aggressive or high-risk. Of course, with high risk, the return expected by financiers is also relatively high. Financiers must be able to measure risk and carry out good money management before investing through the *Urun Dana* service. Additionally, the maintenance of the trust of these financiers must also be ensured by the organizer by providing a qualified issuer so that it can provide a return on investment in accordance with the business projections outlined in the issuer's prospectus.

The SCF industry in Indonesia has great potential for development. The very large number of MSMEs in Indonesia provide 61% of Indonesia's GDP. Based on OJK data, until the end of December 2020, the number of issuers/business actors who utilize ECF from four organizers only reached 129 issuers, with the amount of funds raised reaching IDR 191.2 billion. This number is very small when compared to, for example, financing through peer-to-peer lending. When viewed from the positive side, SCF still has room for development. SCF, if it can be developed properly, can be an alternative source of funding for UMK in Indonesia. So far, UMK in Indonesia is still constrained by the financier aspect. The main source of this capital problem is that it is still dominantly sourced from banking institutions.

### 9. The Development of Equity Crowdfunding Arrangements

Funding sources through equity crowdfunding (ECF) are seen as one of the alternatives for micro, small, and medium enterprises to be able to access the source of funding they need. The state must be present in this regard to facilitate the strengthening of activities of MSMEs, considering that micro, small, and medium enterprises are the backbone of economic activity in a country, but they face severe funding constraints in traditional loans and capital markets.

Several countries have responded to the development of ECF by issuing various accommodative and fair regulations to ensure certainty and legal protection for business actors. Generally, each country issues ECF regulations by accommodating the use of digital platforms.

1. The main content of ECF regulation in the UK is to limit the types of investors that companies can send direct offer promotions to unregistered equities or debt securities. Investors who are allowed to participate in public offerings through equity funding in the UK are professional clients; retail clients who confirm that, in connection with the investment being promoted, they will receive regulated investment advice or investment management services from an authorized person; retail clients who are venture capital contacts or corporate financial contacts; retail clients who are certified or self-certified as sophisticated investors; retail clients who are certified as high net worth investors; and retail clients who clarify that they will not invest more than 10% of net financial assets that can be invested in unregistered equity and debt securities. The ECF regulatory framework in the UK is one of the largest financial supervision regulations in the world and the most complex given the scope of its arrangements relating to the integrated financial system including developed securities markets. Even the IMF has noted the importance of the UK equity market as a platform for trading foreign stocks, particularly European stocks. There are two forms of crowd-funding set up in the UK: first, a loan-based crowdfunding platform where investors can lend money to the issuer in the hope of a financial return in the form of interest

payments and capital repayment over time; second, an investment-based crowdfunding platform where investors can invest in unregistered shares or debt securities from the issuer. ECF falls into the category of investment-based crowdfunding in the UK. Both forms of crowdfunding were previously subject to general regulations regarding the lending and issuance of securities. In its development, the British government created a special regulatory regime for crowdfunding. The ECF regulatory framework in the UK is media neutral, meaning it applies to all intermediary marketing offerings for "unreelable securities" (securities not listed on the legal stock market and no secondary market) using either online portals or offline mechanisms. This regulation applies to any ECF platform operator offering services to invest in new and established businesses by buying shares (and debt). The UK ECF regime effectively limits its arrangements to public companies because the rules make no exception to the ban on public offerings of shares by private companies. For a company to be classified as public, it must qualify that it does not have a face value of allocated share capital below GBP 50,000. If a public company wants to issue shares through the ECF platform, it must comply with many of the rules that apply equally to ECF platform operators. Companies issue shares through the ECF platform deal with investments. ECF issuers are not limited to the amount they can collect through the ECF process.

2.  ECF arrangements in Australia are subject to Australian Securities and Investments Commission oversight. ECF platform operators in Australia are referred to as CSF intermediaries. They are considered financial service licensees (and in some cases Australian Market License holders—although holding an Australian Market License will not remove the need to hold a financial services license when providing crowdfunding services) whose license expressly authorizes the licensee to operate the ECF platform. Therefore, the provision of crowdfunding services is the provision of financial services, which requires a financial services operational permit called the Australian Financial Services License (AFSL). ECF platform operators have a number of obligations including not publishing the offer documents presented by the issuer; the provision of risk warnings on ECF platforms in accordance with regulations; offering mechanisms that allow investors to buy shares but only after they complete recognition; offering communication facilities so that people accessing the offer documents through the platform can make posts related to the offer, view other people's posts, and ask questions to ECF platform publishers and operators; and ensuring that the direct or indirect costs and interests of ECF platform operators are clearly displayed on the platform. ECF Regulation Australia covers two broad areas of regulation for issuers: (1) eligibility requirements for determining who can issue shares through the ECF regime, and (2) disclosure of information to issuers raising funds through the ECF regime. Regulations regarding operators and issuers of ECF platforms are published to protect investors. One of them is by setting an upper limit on the amount that can be invested by a particular investor.

3.  The United States enacts the JOBS Act as the basis of ECF regulation by excluding control over public offerings under the Securities Act. More specifically, the rules waive the obligation to submit a registration statement when fulfilling the four elements, namely (1) the offer limit for the issuer, (2) the investment limit for investors, (3) the requirements for the broker or funding portal, and (4) the requirements for the issuer (De Moor and Kim 2016). ECF platform operators must be registered with the Securities and Exchange Commission (SEC) either as broker-dealers or as "funding portals". Funding portals are less strictly regulated than broker-dealers, but they cannot offer investment advice or recommendations; solicit purchases, sales, or offers to purchase securities offered on their ECF platform; compensate employees, agents, or others for solicitation; or withhold, manage, own, or handle investor funds or securities. If prospective ECF platform operators want to engage in any of these activities, then they need to register as broker-dealers (De Moor and Kim 2016, p. 4). The United States also regulates the activities of issuers. First, there is a limit on the types of companies

that can issue shares through ECF. Issuers must be private companies and based in the United States. Issuers eligible to use ECF can raise a maximum aggregate amount of USD 1 million within a 12-month period. This amount is adjusted for inflation at least once every 5 years. In addition to these financial constraints, the issuer must fulfill several obligations. For example, once an issuer collects funds through the ECF, they must submit bid information to the SEC, and they may have to file an annual report with the SEC. Issuers must electronically submit their offer statement on Form C through the SEC's Electronic Data Collection, Analysis, and Retrieval (EDGAR) system and with the relevant ECF platform operator (De Moor and Kim 2016, p. 5).

4.  Malaysia enforces comprehensive ECF regulations as the basis for equity regulation. The regulation aims to facilitate the development of crowdfunding potential to facilitate innovation, productivity, and growth by encouraging the creative potential of SMEs; and the potential for increased competition among capital suppliers that can lower capital costs for all issuers. To that end, the Malaysian government issued Guidelines on Recognized Markets as the basis of implementing regulations to establish the operation of the ECF. There are at least 25 ECF issuers, with a total of RM14 million collected by these issuers. ECF issuer industries that have successfully operated in Malaysia include industries in the education, retail, digital human resources, and e-commerce sectors (ASEAN 2017, pp. 37–38). At its broadest level, the ECF market guidelines set out two requirements for ECF platform operators. The first relates to the registration of ECF platform operators, and the second relates to their ongoing regulations once registered. Guidelines establishing requirements regarding the registration of market operators which include orderly, fair, and transparent operation of the market through its electronic facilities; responsibility for proper operation; management of risks associated with the business and its operations; market rules that provide investor protection and the public interest; assurance of the proper functioning of the market; the promotion of fairness and transparency; management of any conflicts of interest that may arise; the promotion of fair treatment of its users or anyone who subscribes to its services; the promotion of the fair treatment of any person hosted, or applicable to on its hosting, on its platform; assurance of proper arrangement and supervision of its users, or any persons using or accessing its platform, including the suspension and expulsion of such persons; and the provision of legal appeals (ASEAN 2017, pp. 39–40). Each ECF platform operator must comply with the obligation (a) to conduct due diligence on prospective issuers who plan to use its platform. Due diligence includes background checks on issuers to ensure they, the board of directors, senior management, and controlling owners are in good health and proper, and verifying the issuer's proposed business; (b) monitoring and ensuring compliance with its rules; (c) implementing investor education programs; (d) ensuring that issuer disclosure documents submitted to ECF operators are verified for accuracy and accessible to investors through the platform; (e) notification of investors of any adverse material changes to the issuer's proposal; (f) assurance that the fundraising limits imposed on the issuer are not breached; (g) assurance that investment restrictions imposed on investors are not breached; (h) obtainment and storage of self-declared risk recognition forms from investors before they invest in the ECF platform; and (i) the possession of a flow monitoring process mechanism to monitor anti-money laundering requirements. In addition, the guidelines also govern that information disclosed by prospective issuers must be accurate and must not contain untrue or misleading statements. If the operator of the ECF platform discloses information containing false or misleading statements, it may incur a fine or imprisonment for a period of no more than 10 years, or both. Finally, the guidelines set rules regarding the handling of trust funds by ECF platform operators. ECF platform operators must create and maintain one or more underwriting accounts at ECF's licensed hosting agency where funds are collected by the issuer. This underwriting money can only be issued by the ECF platform operator to the issuer after a predetermined amount

has been reached. ECF Malaysia regulations also set limits on the types of companies as issuers that can raise funds by issuing common shares (common shares and pre-ferred shares) through the ECF and how they can use the system. ECF in Malaysia is restricted to private limited liability companies incorporated in Malaysia. Companies wishing to use ECF can have more than 50 non-employee shareholders by using a nominee structure for shareholders. Issuers must also comply with several require-ments regarding the disclosure of information. Before issuers can be hosted on the ECF platform, they must provide the following information to the ECF platform operator (at a minimum) including information that explains the company's key characteristics; information that describes the purpose of the fundraiser and the amount the publisher wants to collect; information relating to the company's business plan; and financial information related to the company. If the publisher advertises an offer, the ad may not contain suggestions that may trigger a requirement to obtain a Capital Market Service License. This license is required in Malaysia to provide services regarding, among other things, securities transactions, investment advice, and financial planning. The many regulations for ECF platform operators and issuers issued by the Malaysian government are efforts aimed at protecting investors. For example, requirements regarding disclosure and conflicts of interest are imposed on operators and issuers of ECF platforms for the benefit of investors.

5.   Thailand also regulates the ECF, which includes requirements for the registration of the ECF platform and the continued implementation of the ECF platform. Regarding the list requirements, prospective ECF platform operators must apply to Thailand's financial services authority and pay the relevant application fee. Prospective platform operators must be established under Thai law, have registered capital of no less than THB 5 million, have no financial difficulties or other deficiencies that would make them unsuitable as ECF platform operators, have organs consisting of directors and administrators who are not prohibited from participating in capital market business activities, and have the capacity and capability to implement the system necessary to operate the platform according to certain requirements set out in ECF regulations (ASEAN 2017, pp. 45–46). The ECF rule regime in Thailand places legal responsibility on ECF platform operators to ensure investor protection. Therefore, ECF platform operators must act honestly, have due diligence, and have no conflicts of interest when carrying out their functions in accordance with ECF regulations. The ECF arrangement in Thailand classifies the types of issuers into two types of corporate companies: public limited liability companies and limited liability companies. A public limited liability company may offer shares for sale to the public or to anyone if the offer is made in accordance with relevant laws regarding securities and stock exchanges. A limited liability company is a closed company that is prohibited from ordering shares to the public. Both types of companies are allowed to issue shares through ECF if the company is established under Thai law, have a business plan and intend to conduct business with funds raised through the ECF platform, and never issue shares through private placements or public offerings. It is not allowed to list its shares on the Thai Stock Exchange. Thailand's ECF regulations expressly seek to protect investors, including restrictions on the amount that can be invested depending on the investor category, requirements regarding self-accreditation, and educational requirements. There are two categories of investors recognized in Thailand's ECF regulations: retail investors and non-retail investors, including: (1) institutional investors, (2) private equity trusts, (3) venture capital businesses, and (4) qualified investors.

## 10. Design of Indonesian Equity Crowdfunding Arrangement That Can Provide Guaranteed Legal Certainty in the Digital Age

The development of the business world based on information technology has grown very rapidly. This very rapid development not only occurs in the real sector but also in the financial sector. According to the Chairman of the Board of Commissioners, Wimboh

Santoso mentioned that financial technology in Indonesia is developing tremendously because it can reach remote areas. It is estimated that P2P lending fintech services in Indonesia have reached 5,160,120 customers, where there was an approximate increase of 18.91% in January 2019 (Agustina et al. 2019, p. 21).

The development of the business world based on information technology has given birth to various new business activities and new regulations related to the new business activities. One of them, regarding business activities in the financial sector, that is currently a trend is equity crowdfunding business activities. Equity crowdfunding business activities are business activities in the financial sector that utilize digital platforms to be able to raise funds from the public, also known as investors, and distribute them to people who need these funds, also known as debtors, for a business activity development plan from debtors. The distribution of these funds is not realized in the form of money loans but is realized in the form of capital participation (shares) or in the form of debentures.

To be able to smooth this business activity, the regulator, in this case, OJK, has issued OJK regulations. The regulation in question is The Financial Services Authority Regulation No. 16/POJK.04/2021 on Changes to Financial Services Authority Regulation No. 57/POJK.04.2021 on Securities Offerings Through Information-Technology-Based Fund *Urun* Services.

However, this arrangement is considered inadequate and has not been able to guarantee legal certainty for the parties involved in these business activities. This is as identified in the context of the first problem discussion of this study. Furthermore, after identifying the problems arising from the OJK arrangements available and by paying attention to arrangements in other countries, it can finally be understood that there is a need for more adequate arrangements. To be able to provide the right design arrangement, it is necessary to understand the legal design that should be provided regarding the intersection between law and technology.

1.  Legal and Technological Intersex Arrangement Design

In many literature designing settings that have an intersex relationship with technology, then there are several design settings that can be used. The design of such arrangements includes (1) government-regulation-based arrangement design and (2) the design of a self-regulation-based arrangement.

Government regulation is a form of regulation enacted by the government to regulate various human activities. In this context, the interaction between actors on the Internet should not be separated from government regulation.

Government regulations already regulate behaviour, adjudicate disputes, and provide solutions to mistakes made in cyberspace. Before adding a new level of governance to cyberspace, the government must first determine whether existing government regulations are sufficient to protect the government's sovereign interests in regulating cyberspace. The interests of the government must be measured by the externalities of the actions in cyberspace. As the effects of cyberian actions or inaction become greater in the real world, the government's interest in regulating actions experiences inaction. This is implied in the self-regulation model that regulates more in cyberspace (Gibbons 1997).

Meanwhile, the design of self-regulation-based arrangements is a unique arrangement that combines private interests and government supervision, which is an effective and efficient form of regulation for the complex and dynamic financial services industry. The main purpose of self-regulation is the same as government regulation, which is to maintain market integrity (a fair, efficient, and transparent market), preserve financial integrity (reduce systemic risks), and protect investors (Report of the SRO Consultative Committee of the International Organization of Securities Commissions 2000, p. 1).

Some of the essential elements for self-regulation can work effectively, so it must contain the following elements (Report of the SRO Consultative Committee of the International Organization of Securities Commissions 2000, pp. 4–7):

a. Industry specialized knowledge

Financial markets are becoming increasingly complex. SRO has an in-depth interest and knowledge of the industry and the regulatory framework within which they operate. In an environment characterized by a variety of different markets and different types of participants, specialized and thorough knowledge is very beneficial. It is an invaluable source of expertise that can be used by legal regulators and provides several benefits. For example, SRO and its members should be involved in all development discussion rules that affect their industry. SRO typically adopts, updates, and enforces its own procedural rules and rules of conduct, using a large voluntary market network of professionals to provide hands-on market experience. As such, SRO has the expertise and direct market contacts necessary to stay abreast of rapid changes in complex industries and continue regulatory effectiveness. This experience can greatly improve the effectiveness and efficiency of SRO's program of the development, monitoring, and enforcement of rules.

b. Industry motivation

Self-regulation systems work because of business incentives to operate fair, financially sound, and competitive markets. Reputation and competition are strong motivational forces for maintaining appropriate behaviour, especially in today's global environment, where market participants have almost direct, 24-h access to a variety of competing markets and products. To realize the concept of "self" in self-regulation, SRO must encourage market participants and their professional trade associations to contribute to the development of industry best practices and standards.

c. Contractual relationship

The contractual relationships that an SRO has with the individuals and entities it regulates can be very strong. It can have a global reach, crossing national boundaries; contractual relationships also provide more flexibility and allow SRO to react more quickly as it is based on the SRO rulebook and an agreement by SRO members that they will comply with the requirements of the SRO rulebook. The process of revising an SRO contract agreement can be a less complicated process than having to change legislation through legislative action.

d. Transparency and accountability

SRO compliance programs must be transparent and accountable to ensure that SRO follows standards of professional conduct in matters including confidentiality and procedural fairness. Such transparency can occur in different ways, including making SRO rules accessible to the public in print or over the Internet, publicizing significant disciplines of action taken by SRO, and through educational outreach programs. The inclusion of both public representatives and industry professionals in the SRO governing body and public participation in deliberations relating to regulatory policy and rulemaking can also provide a foundation for open organizations. In some jurisdictions, SRO prepares regulatory plans that are submitted to their legal regulators and made available to the public. This regulatory plan describes the purpose of SRO regulation, what the SRO wants to do in the next year, how it will do it, and how much it will cost. The SRO plan should consider a cost–benefit analysis.

e. Flexible SRO compliance program

Self-regulation allows for more diversity in methods of compliance with rules and regulations than legal regulators might provide. The regulatory framework must be flexible enough to enable market participants to respond to inevitable changes in innovative, timely, and sensitive ways.

f. Coordination and information sharing

As the market becomes more globalized, market coordination of surveillance becomes more important. Coordination does occur between SRO and SRO and regulators. SRO is an excellent forum for bringing together various interests in the regulation of issues. With the elimination or reduction of the cap, regulatory arbitration that regulated parties

may be interested in a cheaper regulatory environment could become more common. An identical approach among international regulators, however, is an unrealistic goal. Although instruments regulated by different international regulators may be very similar, each country's priorities and culture are not. Coordination and information sharing should be a priority among markets to address cross-market issues. A coordinated approach is a needed to address potential market abuse or systemic risk issues that can impact more than one market.

2.  Co-Regulation as Equity Crowdfunding Arrangement Design Provides Guarantee of Legal Certainty for Parties

Through understanding the design of government regulation and self-regulation-based arrangements in relation to equity crowdfunding arrangements implemented in Indonesia, the regulatory model can be divided into two types, namely self-regulation and government regulation.

The design of government-regulation-based equity crowdfunding arrangements is based on the authority of financial services institutions in shaping rules in the field of financial services. This authority is granted by legislation related to the financial services authority itself. The nature of the rules made by financial services authority institutions is binding in general to financial service actors. In addition to having binding powers in general, the design of this arrangement usually has sanctions in the form of administrative, civil to criminal sanctions against financial service actors who violate the provisions, and sanctions can be imposed.[30]

Currently, the design of government-regulation-based equity crowdfunding arrangements can be seen in the provisions of Financial Services Authority Regulation No. 16/POJK.04/2021 on Changes to Financial Services Authority Regulation No. 57/POJK.04.2021 on Securities Offerings Through Information-Technology-Based *Urun Dana* Services. The regulated aspects include general provisions, fund *Urun* service organizers, *Urun Dana* services, users of *Urun Dana* service services, *Urun Dana* service agreements, risk mitigation, education, and protection of users of *Urun Dana* services, the principle of knowing customers, other provisions, administrative sanctions, transitional provisions, and closing provisions.

Meanwhile, self-regulation-based equity crowdfunding arrangements are regulated by actors who organize equity crowdfunding activities. This is by equity crowdfunding institutions such as *PT. Investment Digital Nusantara, PT CrowdDana Teknologi Indonusa, PT Dana Saham Bersama, PT Dana Investasi Bersama*, and several others.

Self-regulation can be found on every website service of equity crowdfunding organizers contained in the terms and conditions feature. Aspects stipulated in the terms and conditions as regulated by *PT. CrowdDana Teknologi Indonesia* include definitions, user services, accounts and passwords, contacts, copyrights and trademarks, the registration process, account balance, account balance risk, ban, withdrawal of funds, securities offerings, service fees and other fees, service implementation agreements, links to other websites, viruses, hackers and other violations, cookies, user disputes, adverse actions, changes, disclaimers, complaints of disputes, and applicable law.

Looking at this reality, the Indonesia is reasonable in regulating equity crowdfunding implementing co-regulation design. Co-regulation design is a regulatory approach that emphasizes the sharing of responsibilities between countries and non-state actors such as broad-based private sector stakeholders in policy making and implementation. It focuses on collaboration in the creation, adoption, enforcement, and evolution of policies and regulations. It is helpful to regulate the digital economy because it can provide countries with the necessary data and knowledge, mechanisms for dialogue, and flexible adaptation of legislative solutions in a new and rapidly changing digital economy and facilitate implementation regulation.

---

[30] *Provisions of Chapter III, Objectives, Functions, Duties, and Authority of Law No. 21 of 2011 concerning Financial Services Authority*, 2011.

For the design of this co-regulation-based crowdfunding arrangement that can provide guarantees of legal certainty to the parties, several things must be strengthened, including aspects of government regulation and self-regulation. From the aspect of government regulation, there are two things that must be strengthened: container and substance of government regulation. This must be strengthened from the aspect of this government regulation container in the form of strengthening the design of equity crowdfunding arrangements from the form of financial services authority regulations into law. The design of the arrangement in the form of legislation includes all forms of finance technology activities, one of which regulates equity crowdfunding. Through strengthening aspects of government regulation, the rules will be more open and can involve optimal participation from stakeholders and have a stronger position than the design of the arrangements available today. An aspect in terms of the substance of government regulation that must be strengthened is the scope of regulation, governance, institutional, enforcement procedures, and sanctions that can be imposed.

The scope of the arrangement will include aspects of growing finance technology, including equity crowdfunding. Governance emphasizes the aspects of coaching, implementation, and standardization in the preparation of the policies of each finance technology service, the management and supervision of every form of financial technology, and synchronization between finance technology if they have a relationship with each other. Institutions are pressed on the importance of the specific duties and authorities of financial services authority institutions in controlling the management of financial technology based on the principles of good corporate governance. Enforcement procedures and sanctions are expected to be insufficient with generally applicable law enforcement procedures but still accommodate specific law enforcement procedures such as through online dispute resolution. Meanwhile, in terms of the implementation of sanctions developed, a pattern of imposition of sanctions that prioritizes aspects of economic loss recovery considering this rule is closely related to economic activity.

## 11. Conclusions

Equity crowdfunding (ECF) arrangements in Indonesia have undergone two changes since first being promulgated in Financial Services Authority (POJK) Regulation No. 37/2018. POJK No. 57/2020 Junto POJK No. 16/2021 is the only legal umbrella that regulates the existence of the ECF service industry in Indonesia today. In substance, the arrangement regarding ECF services should provide legal protection and certainty for the parties in ECF, namely organizers, issuers, and financiers. However, the current POJK is still not enough to provide protection and legal certainty, especially to financiers of ECF services. This happens because, in terms of regulatory substance, it is not sufficient because it is only regulated in OJK regulations and, in terms of regulatory enforcement and supervision, by regulators who are still weak. Of the three parties in the service of ECF, financiers are the most vulnerable to risks in the service industry. Regulators should be involving financiers in policy making and the regulation of funds so that regulations can be created that can provide legal certainty for all parties in the service of ECF in Indonesia.

Many countries are developing equity crowdfunding regulatory frameworks oriented towards investor protection. Regulations developed in the digital age place more emphasis on controlling ECF platform operators who must comply with a number of licensing or registration requirements and ensure they are suitable to provide services and must comply with a number of ongoing obligations for the platform to operate in an appropriate manner. In addition, the development of this regulation also emphasizes the compliance of issuers in terms of the disclosure of information. In addition, protection for investors is ensured by containing regulations that limit the amount that can be invested and ensure that investors have the necessary information.

The design of equity crowdfunding arrangements that provide guarantees of legal certainty to the parties can be made in the form of co-regulation arrangements. Co-regulation design is a regulatory approach that emphasizes the sharing of responsibilities between

countries and non-state actors such as broad-based private-sector stakeholders in policy making and implementation. It focuses on collaboration in the creation, adoption, enforcement, and evolution of policies and regulations. It is helpful to regulate the digital economy because it can provide countries with the necessary data and knowledge, mechanisms for dialogue, and flexible adaptation of legislative solutions in a new and rapidly changing digital economy and facilitate implementing regulations.

**Author Contributions:** Formal analysis, L.S.W.; writing—original draft, B.A.R.; writing—review and editing, A.A. All authors have read and agreed to the published version of the manuscript.

**Funding:** This research received Faculty of Law, Universitas Islam Indonesia funding.

**Institutional Review Board Statement:** Not applicable.

**Informed Consent Statement:** Not applicable.

**Data Availability Statement:** Not applicable.

**Conflicts of Interest:** The author declares no conflict of interest.

## References

### *Primary Sources*

https://bizhare.id/, accessed on 1 June 2020.

Undang-undang Nomor 21 Tahun 2011 tentang Otoritas Jasa Keuangan (Law Number 21 Year 2011 On Financial Services Authority).

(POJK.04/2021) Peraturan Otoritas Jasa Keuangan (POJK) No. 16/POJK.04/2021 tentang Perubahan Atas Peraturan Otoritas Jasa Keuangan (POJK) Nomor 57/POJK.04.2020 tentang Penawaran Efek Melalui Layanan Urun Dana Berbasis Teknologi Informasi (Regulation of Financial Services Authority Number 16/POJK.04/2021 Concerning Amendment to Financial Services Authority Regulation (POJK) Number 57/POJK.04.2021 Concerning Securities Offering Through Information Technology-Based Crowdfunding Services)

(POJK.04/2020) Peraturan Otoritas Jasa Keuangan (POJK) No. 57/POJK.04/2020 tentang Penawaran Efek melalui Layanan Urun Dana Berbasis Teknologi Informasi. (Regulation of Financial Services Authority Number 57/POJK.04/2020 Concerning Securities Offering through Information Technology-Based Crowdfunding Services)

(POJK.04/2018) Peraturan Otoritas Jasa Keuangan (POJK) No. 37/POJK.04/2018 tentang Layanan Urun Dana Melalui Penawaran Saham Berbasis Teknologi Informasi (Regulation of Financial Services Authority Number 37/POJK.04/2018 Concerning Crowdfunding Service through Information Technology-Based Share Offering)

### *Second Sources*

Abdurrahman, Muslan. 2009. *Sosiologi dan Metode Penelitian Hukum*. Malang: UMM Press.

Agustina, Lidya, Yan Andriariza Ambhita Sukma, D. Mahmudah, A. B. Setiawan, R. Mustika, A. Dunan, and A. Ratnawati. 2019. *Perkembangan Ekonomi Digital di Indonesia Strategi dan Sektor Potensial*. Jakarta: Puslitbang Aptika dan IKP.

Ahlers, Gerrit, Douglas Cumming, Christina Günther, and Denis Schweizer. 2013. *Signaling in Equity Crowdfunding*. Chicago: Northwestern Law.

Al-Fatih, Sholahuddin, and Zaka Aditya. 2019. Hoax and The Principle of Legal Certainty in Indonesian Legal System. Paper presented at the 1st International Conference on Business, Law and Pedagogy, Sidoarjo, Indonesia, February 13–15.

ASEAN. 2017. *Facilitating Equity Crowdfunding in the ASEAN Region*. Jakarta: ASEAN Secretariat.

Asikin, Amiruddin dan Zainal. 2006. *Pengantar Metode Penelitian Hukum*. Jakarta: PT. Raja Grafindo Persada.

Cicchiello, Antonella Francesca, and Daniele Leone. 2020. Encouraging investment in SMEs through equity-based crowdfunding. *International Journal of Globalisation and Small Business* 11: 258–78. [CrossRef]

Cicchiello, Antonella Francesca, Maria Cristina Pietronudo, Daniele Leone, and Andrea Caporuscio. 2020. Entrepreneurial dynamics and investor-oriented approaches for regulating the equity-based crowdfunding. *Journal of Entrepreneurship and Public Policy* 10: 235–60. [CrossRef]

De Moor, Lieven, and Hyonsu Kim. 2016. A Comparative Study on the Regulatory Framework of Crowdfunding. *The Journal of Small Business Innovation* 19: 1–16.

Dresner, Steven. 2014. *Crowdfunding: A Guide to Raising Capital on the Internet*. Hoboken: John Wiley & Sons, Limited, p. 3. Available online: https://onlinelibrary.wiley.com/doi/book/10.1002/9781118746974 (accessed on 23 December 2021).

Eko, Lyombe. 2013. Internet Law and Regulation. In *The International Encyclopedia of Communication*. Edited by Wolfgang Donsbach. Africa: Blackwell Publishing.

Fahrimal, Yuhdi. 2018. Netiquette: Etika Jejaring Sosial Generasi Milenial dalam Media Sosial Netiquette: The Ethics of Millenial-Generation Social Networks in Social Media. *Jurnal Penelitian Pers dan Komunikasi Pembangunan* 22: 61. [CrossRef]

Financial Conduct Authority Policy Statement. n.d. The FCA's regulatory Approach to Crowdfunding Over the Internet, and the Promotion of Non-Readily Realisable Securities by Other Media. Available online: https://www.sec.gov/divisions/marketreg/tmjobsact-crowdfundingintermediariesfaq.html (accessed on 15 November 2021).

Gabison, Garry A. 2015. *Understanding Crowdfunding and Its Regulations, How Can Crowdfunding Help ICT Innovation*. European Commission Joint Research Centre Science and Policy Report. Luxembourg: Publications Office of the European Union.

Gibbons, Llewellyn Joseph. 1997. No Regulation, Government Regulation, or Self- Regulation: Social Enforcement or Social Contracting for Governance in Cyberspace. *Cornell Journal of Law and Public Policy* 6: 483.

Gray, Christoper Berry, ed. 1999. *The Philosophy of Law an Encyclopedia*. New York and London: Garland Publishing.

Hartanto, Ratna. 2020. Hubungan Hukum Para Pihak Dalam Layanan Urun Dana Melalui Penawaran Saham Berbasis Teknologi Informasi. *Jurnal Hukum Ius Quia Iustum FH UII* 27: 157. [CrossRef]

Hemer, Joachim. 2011. *A Snapshot on Crowdfunding*. Karlsruhe: Franhoufer ISI.

Hossain, Mokter, and Gospel Onyema Oparaocha. 2017. Crowdfunding: Motives, Definition, Typology and Ethical Challenges. *Entrepreneurship Research Journal* 7: 14. [CrossRef]

International Organization of Securities Commissions. 2017. IOSCO Research Report On Financial Technologies (Fintech). Available online: https://www.iosco.org/library/pubdocs/pdf/IOSCOPD554.pdf (accessed on 6 August 2021).

Ishak, Ajub. 2013. Ciri-ciri Pendekatan Sosiologi dan Sejarah dalam Mengkaji Hukum Islam. *Al Mizan* 9: 72.

Jones, Lucinda. 2000. An Artist's Entry Into Cyberspace: Intellectual Property on the Internet. *European Intellectual Property Review* 22: 79–92.

Josua, Gustaf, Moch Zairul Alam, and Ranitya Ganindha. 2019. *Tinjauan Yuridis Terhadap Penerapan Prinsip Keterbukaan Dan Mitigasi Resiko Pada Equity Crowdfunding Di Indonesia*. Malang: Jurnal Hukum Fakultas Hukum Universitas Brawijaya.

Lucchi, Nicola. n.d. Intellectual Property Rights in Digital Media: A comparative Analysis of Legal Protection, Technological Measures, and New Business Models under EU and U.S. Law. *Buffalo Law Review* 53: 118. [CrossRef]

Martin, Michael. 1965. Roscoe Pound's Philosophy of Law. *ARSP: Archiv Für Rechts- Und Sozialphilosophie/Archives for Philosophy of Law and Social Philosophy* 51: 37–55.

Müllerleile, Thomas, Dieter William Joenssen, and Andreas Müllerleile. 2014. Crisis, Co-Financing and Crowdfunding: Igniting Regional Development. Paper presented at the International Conference for Entrepreneurship, Innovation and Regional Development ICEIRD, Nicosia, Cyprus, June 5–6.

Paunio, Elina. 2009. Beyond Predictability—Reflections on Legal Certainty and The Discourse Theory of Law in the EU Legal Order. *German Law Journal* 10: 1469–93. [CrossRef]

Report of the SRO Consultative Committee of the International Organization of Securities Commissions. 2000. Model for Effective Regulation. Available online: https://www.iosco.org/library/pubdocs/pdf/IOSCOPD110.pdf (accessed on 26 December 2021).

Rosana, Ellya. 2013. Hukum dan Perkembangan Masyarakat. *Jurnal TAPIs* 9: 109.

Segura-Serrano, Antonio. 2006. Internet Regulation and the Role of International Law. *Max Planck Yearbook of United Nation Law* 10: 193–201. [CrossRef]

Shaham, Ron. 1995. Custom, Islamic Law, and Statutory Legislation: Marriage Registration and Minimum Age at Marriage in the Egyptian Shari'a Courts. *Islamic Law and Society* 2: 258. [CrossRef]

Soekanto, Soerjono. 2013. *Pokok-Pokok Sosiologi Hukum*. Jakarta: RajaGrafindo Persada.

ter Borg, S. J. A., and W. S. R. Stoter. 2010. The Rule of Law in Comparative Perspective. *IUS Gentium* 3: 4.

Wade, Henry William Rawson. 1941. The Concept of Legal Certainty a Preliminary Skirmish. *The Modern Law Review* 4: 185. [CrossRef]

