# Peer review of "Design of Equity Crowdfunding in the Digital Age"

_laws, 2022_

Round 1

Reviewer 1 Report

I recommend the authors broaden the discussion in the introduction highlighting the contribution of this study. Prior studies have already investigated the aspect of regulation in equity crowdfunding markets. Indeed, this study refers to a different geographic context, which is under-researched. However, in order to make a sophisticated contribution, the study should go beyond a different geographic setting. 

The most relevant literature on the topic at stake is contained in the paper. Yet, I recommend authors to include other relevant papers that can help give more support to their discussion and analysis.

- Cicchiello, A. F., & Leone, D. (2020). Encouraging investment in SMEs through equity-based crowdfunding. International Journal of Globalisation and Small Business, 11(3), 258-278. https://doi.org/10.1504/IJGSB.2020.109553

- Cicchiello, A. F., Pietronudo, M. C., Leone, D., & Caporuscio, A. (2020). Entrepreneurial dynamics and investor-oriented approaches for regulating the equity-based crowdfunding. Journal of Entrepreneurship and Public Policy, 10(2), 235-260. https://doi.org/10.1108/JEPP-03-2019-0010

I also think more could have been made of the discussion on the implications for theorising about the topic. The conclusion needs to go beyond the immediate results and actually explore the ramifications of the research findings for knowledge production and practice in the area.

Authors should pay more attention to the clarity of expression and readability. Some of the sentence construction needs to be clearer.

Author Response

Dear reviewer, 

Thank you for your review notes. We add the description in the background of the paper: 

"In addition, A number of countries in Europe, particularly the UK, Germany, and France, have enacted regulations that do not restrict investors but rather protect and promote them, which has come to be called investor protection-oriented policies. As a result, it is natural that these countries have a record high volume of investment from the public. However, there are also some other countries in Europe, such as Spain and Italy, that apply the opposite approach, namely the restrictive approach, where through this approach, investors are over-protected, causing them to be reluctant to invest.[2]

The phenomenon of state policy responses to the rapid development of the crowdfunding industry indirectly, whether we like it or not, shows that the state has no other choice in responding to the crowdfunding industry but to respond to the challenges in the field of entrepreneurial finance proportionally and ensure protection for investors. This then does not rule out the possibility of giving rise to two forms of domestic regimes with tightening and strengthening regulations; and financial regimes with an orientation to adjust to the needs of the crowdfunding market and local investors. Therefore, a country's government must be able to encourage the growth of crowdfunding by implementing integrity policies that cover the guarantee of internal market stability and facilitate the creation of competitive market conduciveness. Thus, it is hoped that the minimization of information asymmetry and increased transparency and investor protection can be created.[3]

[1] "https://republika.co.id/berita/q9bi9n370/dua-fintech-Urun-dana-ini-kerja-sama-dengan-ksei," accessed June 19, 2020, https://republika.co.id/berita/q9bi9n370/dua-fintech-Urun-dana-ini-kerja-sama-dengan-ksei.

[2] Cicchiello, A. F., Pietronudo, M. C., Leone, D., & Caporuscio, A. (2020). Entrepreneurial dynamics and investor-oriented approaches for regulating the equity-based crowdfunding. Journal of Entrepreneurship and Public Policy, 10(2), 235-260. https://doi.org/10.1108/JEPP-03-2019-0010

[3] Cicchiello, A. F., & Leone, D. (2020). Encouraging investment in SMEs through equity-based crowdfunding. International Journal of Globalisation and Small Business, 11(3), 258-278. https://doi.org/10.1504/IJGSB.2020.109553

Thank you.

Reviewer 2 Report

This paper was interesting and has great potential. Some part of it, mainly the structure, need more work to allow the argument and thoughts presented in this article to shine. 

I would recommend in term of structure:

- Start with intro and make it clear at the end the purpose of the paper and the structure of the paper.

- Law in Indonesia pertaining to equity crowd funding and explain evolution through critical analysis. bring and weave through this part, when appropriate the info in the parts entitled 'library review' and theory of legal certainty' and 'legal theory as a tool of social engineering'.

- have a part on how Indonesian law compares with other jurisdiction. But make sure when doing that you use the information to build toward your argument

- and than the piece de resistance 'Design of Indonesian Equity Crowdfunding arrangement that can provide guaranteed legal certainty in the Digital Age.

- conclusion 

I also recommend that the paper is edited.

Author Response

Dear reviewer, 

Thank you for your review notes. We have adjusted the paper on the basis of your review. Thank you. 

Warm regards

Round 2

Reviewer 1 Report

Dear Authors, 

Many thanks for offering a revised version of your article. Several improvements have been made and my comments are clearly addressed. In my opinion, it is ready for publication after final proofreading.  

Congratulations for this work.